# miRNA Expression Signatures Induced by Chicken Astrovirus Infection in Chickens

**DOI:** 10.3390/ijms242015128

**Published:** 2023-10-12

**Authors:** Joanna Sajewicz-Krukowska, Paweł Mirosław, Jan P. Jastrzębski, Katarzyna Domańska-Blicharz, Karolina Tarasiuk, Barbara Marzec-Kotarska

**Affiliations:** 1Department of Poultry Diseases, National Veterinary Research Institute, 24-100 Puławy, Poland; domanska@piwet.pulawy.pl (K.D.-B.); karolina.tarasiuk@piwet.pulawy.pl (K.T.); 2Foundation of Research and Science Development, 01-242 Warsaw, Poland; pawel_miroslaw@yahoo.com; 3Department of Plant Physiology, Genetics and Biotechnology, Faculty of Biology and Biotechnology, University of Warmia and Mazury in Olsztyn, 10-719 Olsztyn, Poland; bioinformatyka@gmail.com; 4Department of Clinical Pathomorphology, The Medical University of Lublin, 20-090 Lublin, Poland; barbara.marzec@umlub.pl

**Keywords:** chicken astrovirus, white chicks syndrome, microRNA, RNA-seq, spleen, molecular pathogenesis

## Abstract

miRNAs represent ubiquitous regulators of gene expression and play an important and pivotal regulatory role in viral disease pathogenesis and virus–host interactions. Although previous studies have provided basic data for understanding the role of miRNAs in the molecular mechanisms of viral infection in birds, the role of miRNAs in the regulation of host responses to chicken astrovirus (CAstV) infection in chickens is not yet understood. In our study, we applied next-generation sequencing to profile miRNA expression in CAstV-infected chickens and to decipher miRNA-targeted specific signaling pathways engaged in potentially vital virus-infection biological processes. Among the 1354 detected miRNAs, we identified 58 mature miRNAs that were significantly differentially expressed in infected birds. Target prediction resulted in 4741 target genes. GO and KEGG pathway enrichment analyses showed that the target genes were mainly involved in the regulation of cellular processes and immune responses.

## 1. Introduction

Chicken astroviruses (CAstV) are omnipresent single-stranded RNA viruses. Their incidence, reported in multiple countries such as Germany, China, Finland and Poland, is of clinical and economic importance since they are responsible for many diseases [1,2]. The virus shows tropism to intestinal epithelial cells, leading to its increased permeability, resulting in diarrhea. In chickens, multiple CAstV-associated enteritis and clinical symptoms were reported. Extra-intestinal involvement has also been observed, such as runting–stunting syndrome, death caused by nephropathy and visceral gout, or mid-to-late dead-in-shell embryos leading to decreased hatching. Relatively recently, CAstV has also been implicated as the causal factor of “white chick syndrome” (WCS), affecting broiler chicks associated with the white plumage of hatched chicks [3,4,5]. Over the years, of great concern is the increasing severity of observed symptoms, probably due to recombination events and the accumulation of point mutations which may contribute to the genetic variation [6].

Astrovirus, similar to any other virus, elicits an immune response upon entering the host cell, leading to its replication and dissemination [7]. Innate and adaptive signaling pathways are induced to prevent or limit viral invasion. Innate immune responses mediated by Toll-like receptors (TLRs), mainly TLR3 and TLR7-9 and NOD-like receptors (NLRs), contribute to the activation of pro-inflammatory cytokines and interferons (IFNs) which is a triggering event leading to the recruitment of inflammatory cells and adaptive immune response activation [8].

The inflammatory response of the host is always accompanied by dramatic molecular/epigenetic background alteration to modulate the degree of this process and/or prevent viral escape from the control of the immune system. Among a plethora of molecules navigating the immunological response are miRNA [9].

The miRNA, together with other types of non-coding RNA, such as small interfering RNAs, PIWI-interacting RNAs, small nuclear RNAs, small nucleolar RNAs, and long non-coding RNAs, are the most abundant molecules comprising most of the genome [10]. They are relatively primary and sequence conservative, being transcribed both in viruses and humans using the same canonical pathway [9]. They are recognized as key gene expression regulators responsible for the modulation of intercellular signaling and multiple biological processes, both physiological and pathological [10].

Pairing the complementary 3′ Untranslated Region (3′-UTR) in targeted mRNAs, miRNAs trigger their degradation and prevent subsequent protein translation in a tissue- and cell-specific manner [11]. This is only one facet of the overarching regulatory role of miRNA. Viral infection, which is exclusively dependent on the host organism, is also regulated by miRNA. Both viral and host miRNAs are engaged in infectious processes such as virus entry, its replication or the organism’s attempt to confine or modulate the reaction [12]. Jopling et al. performed studies in HCV-infected patients and revealed that host miRNAs showed the potential to interact with the viral genome. miR-122 was confirmed to enable the replication of hepatitis C virus [13]. More recent studies carried out on Severe Acute Respiratory Syndrome Coronavirus 2 (SARS-CoV-2) showed miRNA antiviral defense. Viral protein spike (S), essential for SARS-CoV-2 replication, was successfully downregulated by several human miRNAs such as miR-15a, miR-153, miR-298, miR-508, miR-1909 and miR-3130 [14].

Not only have specific miRNAs involved in the modulation of viral gene expression during infection been characterized, but also a plethora of particular miRNA molecules engaged in human immune response has been described [9]. In mammals, miRNAs were confirmed to be crucial in the development and differentiation of B and T cells, proliferation of monocytes and neutrophils, antibody switching and the release of inflammatory mediators [15]. Data concerning miRNAs’ role in immune response modulation in the veterinary field are still scarce. There are some reports regarding the miRNA profile in Influenza A virus (IAV)- or Gallid herpesvirus 2 (GaHV-2)-infected poultry [16,17]. Leghorn chickens disease ridden with H5N3 virus overexpressed miR-146, which has been previously described as an inhibitor of signaling proteins engaged in the innate immune responses by NF-kappaB [17]. Still, there is a dearth of knowledge regarding astrovirus-induced miRNA-mediated immunological response in infected chickens. In our study, we applied next-generation sequencing (NGS) to profile miRNA expression in CAstV-infected chickens and to decipher miRNA-targeted specific signaling pathways engaged in potentially vital virus-infection biological processes.

## 2. Results

### 2.1. Clinical Features of CAstV-Infected Chickens

No symptoms were observed in virus-infected chickens up to 4 dpi. The spleens showed no apparent differences between the control and infected groups. However, viral RNA in cloacal swabs at 4 dpi has been detected, demonstrating that CAstV replicated in infected chickens.

### 2.2. Analysis of miRNA Expression

Small RNA sequencing yielded 117,275,902 and 119,274,037 reads for the control and infected birds, respectively. The GC content of control was 53.28% and the ratio of bases with a Phred quality score  ≥  30 (Q30) was 95.27%. The GC content of infected chickens was 53.15% and the Q30 was 95.19%. For both the control and infected samples, reads were aligned to the chicken reference genome and miRBase v21 database to classify the known miRNAs.

The differential expression of miRNAs was determined using DESeq2. These results were clearly visualized by clustering the samples by the different treatment (virus infected and non-infected) (Figure 1 and Figure 2) and by constructing an MA plot of the DE miRNAs (Figure 3). miRNA with adjusted *p*-value < 0.05 was selected as significantly differentially expressed. Among the 1354 detected miRNAs, 58 mature miRNAs that were significantly differentially expressed were identified, 39 of which were upregulated and 19 were downregulated in infected birds compared with the controls. The fold changes of the 10 most up- and downregulated differentially expressed miRNAs are presented in Table 1. Among these DE miRNAs, gga-miR-2954 and gga-miR-3533 showed the highest upregulated Log2fold change of 3.22 and 2.83, respectively, while gga-miR-7b and gga-miR-1664-3p showed the highest downregulated Log2 fold change of −9.92 and −1.33, respectively.

### 2.3. Gene Ontology and KEGG Pathway Enrichment Analyses of Target Genes

The target genes of DE miRNAs were predicted using the miRDB. Target prediction resulted in 4741 target genes for 58 mature miRNAs. For the complete lists of target genes and their corresponding miRNAs, see Appendix A. Predicted target genes were used for GO and KEGG pathway enrichment analyses. The Gene Ontology annotations were enriched with target genes and were divided into three groups: biological process (BP), molecular function (MF) and cellular component (CC). Based on the biological process (BP), the target genes were classified into 73 categories, of which top ten enrichment were “regulation from RNA polymerase II promoter”, “intracellular signal transduction”, “regulation of GTPase activity”, “protein phosphorylation”, “axon guidance”, “protein polyubiquitination”, “peptidyl-serine phosphorylation”, “nervous system development”, “positive regulation of synapse assembly” and “semaphorin-plexin signaling pathway”.

In case of molecular function, the terms were classified into 37 categories, with the 10 most enriched involved in “protein/metal ion/chromatin binding”, “transcription factor activity” and “GTPase regulator activity”. Enrichment analysis demonstrated that the cellular component (CC) group of 30 items, the most important of which are nucleus, cytoplasm, synapses, Golgi apparatus, phosphatidylinositol 3-kinase and membrane were correlated with target genes. Table 2, Table 3 and Table 4 show the top 10 significantly enriched GO biological process, molecular function, and cellular component categories. In KEGG pathway enrichment analysis, target genes were found to be enriched in 20 KEGG pathways (*p*-value < 0.05), with the mitogen-activated protein kinase (MAPK) signaling pathway being particularly enriched with the target genes of DE miRNAs, but also the phosphatidylinositol signaling system, endocytosis, autophagy, the adherens junction, the Wnt, FoxO, ErB, insulin and mTor signaling pathways, inositol phosphate metabolism, focal adhesion and others (Table 5). Full lists of enriched GO and KEGG annotations can be found in Appendix A.

### 2.4. Verification of DE miRNAs by qRT-PCRmiRNA

Real-time expression analysis of infected and non-infected chicken was performed to validate the NGS date. The miRNAs of the highest significance, based on NGS data, selected for further analysis included miRNA2954, miRNA3533, miRNA7 and miRNA1664. The validation set was composed of five astrovirus-infected and five non-infected chickens. The real-time miRNA expression confirmed NGS data (Figure 4). Despite the small group, miR-1664 and miR-2954 showed significantly decreased and increased expression, respectively, in astrovirus-infected chickens compared with the healthy ones (*p* = 0.028 and *p* = 0.016) (Table 6). Analysis of two other miRNAs, miR-3533 and miR-7b, revealed a very strong tendency to be overexpressed in the astrovirus-infected cohort (both *p* = 0.057).

## 3. Discussion

In the present study, a next-generation sequencing approach was applied to detect differentially expressed miRNAs in chicken spleens in response to CAstV infection. The spleen is the largest lymphoid organ in birds that contains a large number of immunocompetent cells and can effectively induce innate and adaptive immune responses. Hence, the global profile of miRNA expression in the spleen provided a good overview of the host response to CAstV infection [18,19]. The transcriptional regulation of host miRNAs after CAstV infection in chickens, especially in the spleen, can be used as a tool to study pathogen–host interactions and can therefore provide insights into the pathogenic and immune mechanisms of CAstV.

In our study, a total of 58 known differentially expressed miRNAs were identified in CAstV-infected chickens, including several that may be associated with intracellular or innate immune responses, which are the first line of host defense against virus infection.

miR-2954 expression was described to be significantly altered in the cecum of broilers in response to *Salmonella* infection and probiotic administration. This suggests that miR-2954 may be involved in the immune regulation of the cecum in chickens [20]. The expression of miR-2954 was also significantly upregulated in reticuloendotheliosis virus (REV)-infected chicken embryo fibroblasts [21]. Similarly, through NGS sequencing of small RNA libraries, it has been detected that the expression of miR-2954 in the kidneys of Infectious Bronchitis Virus (IBV)-infected chickens was significantly altered. Based on the results of this study, we also speculate that miR-2954 plays an important role in CAstV–chicken interaction.

Khanduri et al., in their study of the miRNAome and proteome of peripheral blood mononuclear cells of goats infected with the virulent Peste des petits ruminants virus (PPRV), among others, singled out miR-3533 as regulating genes involved in 10 major immune response processes [22].

Expression of gga-miR-1434 was significantly lower in H5N1 avian influenza virus-infected chickens compared to control chickens [21]. These results contrasted with studies by Yang et al. [23] and Guo et al. [24]. In the above, miR-1434 expression was significantly higher in dendritic cells of chickens infected with H9N2 avian influenza virus and fibroblast cells of chicken embryos infected with Newcastle disease virus, respectively. This upregulation of miR-1434 expression was in line with the results of our work.

miR-3535 is involved in the regulation of numerous cellular processes, such as DNA replication, cell cycle progression and DNA damage response. Treatment of Marek’s disease virus (MDV)-transformed T cells with sodium butyrate, an inducer of virus reactivation, significantly increased miR-3535 expression [25].

Decreased expression of miRNA-194 has been demonstrated in Epstein–Barr Virus-infected B cells from patients with post-transplant lymphoproliferative disorder, in turn, overexpression of this miRNA attenuates IL-10 production and increases apoptosis of EBV+ B cell lymphoma lines [26].

Wang et al. demonstrated reduced miRNA-194 expression in A549 alveolar epithelial cells after Influenza A virus IAV/Beijing/501/2009 infection. miR-194 overexpression increased IAV replication by negatively regulating type I interferon (IFN) production. In contrast, miR-194 inhibition attenuated IAV-induced lung injury by promoting type I IFN antiviral activity in vivo. These findings suggest that miR-194 plays an important role in IAV-induced lung injury, and miR-194 antagonism may be a potential therapeutic target during IAV infection [27].

Among miRNAs with an acknowledged role in the veterinary field, the function of a few has not yet been described. In our study, we report for the first time miR-3528, which showed significant upregulation, suggesting its possible important engagement in the host–virus dynamic during infection. To date, there are no data available explaining the exact role of miR-3528 in any biological process.

Peng et al. identified that gga mir-215-5p exhibited significantly varied expression levels of miR-215 between H9N2-infected and non-infected chicken embryo fibroblasts [28]. In addition, miR-215 overexpression increased HCV replication in Con1b cells, while miR-215 silencing inhibited HCV replication in Huh7.5.1 cells [29]. Several literature publications have described differential expression of miR-215-5p in chicken lungs, immune organs and embryonic fibroblasts during H5N3, H5N1 and H9N2 AIV infections and also in chicken kidneys after IBV infection [28,30,31,32].

It was shown that miR-1563 expression was significantly higher in chicken tracheal cells after infection with H4N6 avian influenza virus and chicken kidneys after IBV infection [32,33].

In turn, miR-3536 expression was significantly lower in chicken tracheal cells after infection with H4N6 avian influenza virus [33]. In contrast, miR-3536 expression was significantly higher in Marek’s disease virus-transformed T cell lines compared to control lines [34]. Expression of miR-3536 was strongly upregulated in bursae of Fabricius of chickens infected with very virulent infectious bursal disease virus [35].

Finally, the last of our 10 most upregulated miRNAs, miR-7, is a highly conserved miRNA among different species. MiR-7 has been studied extensively in human organ/tissue development (including brain, pancreas, and thymus) [36], tumor biology (growth, migration, and immune escape) [37], and pathogenesis of diabetes [38]. However, the mechanism of action of miR-7 in antiviral defense has not been well studied. It was shown that a specific siRNA was able not only to effectively inhibit poliovirus (PV) replication, but also to increase miR-7 expression in host cells, which in turn led to increased inhibition of PV infection [39]. In addition, miR-7 targeted and inhibited the expression of the immune factor Myd88 in crabs and was able to affect white spot syndrome virus (WSSV) replication [40]. Zhou et al. found that miR-7 expression significantly increases in human rotavirus infection, and its overexpression of miR-7 inhibits rotavirus replication in vitro [41]. In addition, miR-7 has been shown to increase its expression during influenza virus infection of human respiratory cells, leading to downregulation of antiviral proteins such as Interleukin-1 receptor-associated kinase 1 binding protein (IRAK1) and MAPK3 mitogen-activated protein kinase 3 (MAPK3).

The most downregulated miRNA in our study was found to be miR-7b. It inhibits the translation of Fos protein, which plays an important role in regulating biological processes such as cell proliferation, differentiation, and apoptosis [42]. Increasing miR-7b expression inhibited Pim-1 proto-oncogene (PIM1) expression and initiated apoptosis, in which the host immune system was involved [32]. The fact that miR-7b was the most downregulated miRNA may confirm our previous results suggesting an immunosuppressive effect of CAstV [43].

miR-1664 was found to be the second of the most downregulated miRNAs in this study. Shuo Gao et al., based on the analysis of functional enrichment and clinical manifestations of Reticuloendotheliosis (RE) immunosuppression, selected immune-related target genes that were regulated by gga-miR-1664, among others [44]. This is in line with our previous work and may confirm the immunosuppressive nature of CAstV infection [43]. gga-miR-1664 was also one of the top ten upregulated miRNAs in the study concerning miRNA expression profiling of highly pathogenic avian influenza virus H5N1-infected chicken lungs [45].

As discussed, the 12 differentially expressed (10 up- and 2 downregulated) miRNAs above are likely involved in the regulation of the immune response in CAstV-infected chickens.

The GO analysis showed that the target genes were mainly involved in the regulation of cellular processes and immune responses. One of the most interesting GO terms observed in this study was the regulation of transcription from the RNA polymerase II promoter, indicating that the virus had taken control of the host. The GO analysis revealed a total of 140 terms. It would be difficult to discuss such a multitude of distinct biological processes, molecular functions, and cellular components. Therefore, in this study, we have focused on the analysis of signaling pathways enriched in KEGG analysis, because each of them combines multiple Gene Ontology terms.

Additionally, 20 pathways were found to be enriched in functional KEGG analysis. The most relevant pathways included the MAPK signaling pathway, the phosphatidylinositol signaling system, endocytosis, the adherens junction, autophagy—animal, the Wnt signaling pathway, the FoxO signaling pathway, the ErbB signaling pathway, focal adhesion, the insulin signaling pathway, the mTOR signaling pathway and regulation of the actin cytoskeleton.

Mitogen-activated protein kinase signaling was found to be enriched in the challenger group in KEGG pathway analysis. MAPK cascade is a highly conserved module that is involved in various cellular functions, including cell proliferation, differentiation, and migration, enables the conversion of extracellular signals into cellular responses. In addition, MAPK plays important functions in the immune system [21,46]. It has been proven to be activated in various viral infections in birds, including H5N1 AIV [47,48,49]. It also plays a key role in virus replication in macrophages of chickens infected with H9N2 AIV [50]. Chu et al. showed that NDV V protein, by inducing phosphorylation of ERK protein in the MAPK pathway in host cells, increases viral replication [51]. Moreover, inhibition of ERK activation significantly reduced human astrovirus (HAstV) production in the CAco-2 cell line [52]. This study also indicates that miRNAs that have altered expression likely affect this pathway.

The MAPK signaling pathway was followed by the phosphatidylinositol signaling system. Our results confirm that many viruses activate the phosphatidylinositol 3-kinase (PI3K/Akt) signaling pathway at early stages of infection, leading to increased viral replication. The activated Akt protein inhibits apoptosis while inducing cell survival [53]. Ultimately, it allows the virus to replicate and spread by keeping cells alive. Influenza A virus was proven to increase expression of phosphorylated Akt protein in A549 cells at 6-12 hpi, while its inhibition led to reduced virus entry and replication [54,55]. PI3K/Akt activation was also described for avian leukemia virus and resulted in an increase in virus replication [56]. In addition, Newcastle disease virus further inhibited cell apoptosis and increased autophagy [57,58]. This study also indicates an important role for PI3K in the life cycle of HAstV1 at an early stage of infection, probably during virus entry process [59].

To enter the host cell, most viruses use the mechanism of endocytosis. To date, the entering of the host by enveloped viruses has been widely described in scientific literature, while there is little information on unenveloped viruses, including astroviruses. Entry of unenveloped viruses into the host cell is made difficult by the fact that the hydrophilic virus particle must penetrate the lipid membrane. Endocytic pathways used by various viruses include clathrin-mediated endocytosis, caveolin-mediated uptake, macropinocytosis and novel nonclathrin, non-caveolin pathways that are still poorly characterized [60]. Mendez et al. found that human astroviruses enter Caco-2 cells via a clathrin-dependent endocytic pathway, where they presumably must travel to late endosomes to reach the cytoplasm and initiate the replication cycle [61]. The results of this study also suggest that CAstV uses endocytosis to enter host cells.

Another pathway enriched in this study was the adherens junction. While suggesting a new functional role for IRF7 in AIV infection in chickens, Kim and Zhoe also showed that it regulates genes involved in cell structural integrity or cell assembly. One of these is adherens junctions, which have important functions in the host response to viral infection. Altered expression of proteins involved in adherens junctions can result in increased apoptosis and viral replication [62]. In addition, it was shown that HastV-infected Caco-2 cells had disrupted junctional proteins as early as 6 hpi. [63]. Astrovirus increased barrier permeability in Caco-2 cell culture after apical infection, which correlated with disruption of the tight-junction occludin protein and a reduction in the number of actin stress fibers in the absence of cell death [64].

Autophagy is an evolutionarily conserved process of lysosome self-destruction of harmful components, abnormal cytosolic components, and intracellular pathogens. Autophagy has also been shown to play an important role in antiviral defense. Autophagy recognizes viral particles during infection and limits viral replication [65]. Viruses, in turn, have evolved a variety of mechanisms to avoid or use autophagy for their own survival [66,67]. Literature data indicate that ALV-J negatively regulates autophagy through the GADD45β/MEKK4/P38MAPK signaling pathway, and subsequently mediates apoptosis and promotes viral replication [68,69]. This result suggests that miRNA may be involved in the regulation of CAstV infection through the autophagy pathway. However, the relationships between the target genes and miRNAs, and the specific mechanisms by which miRNAs regulate CAstV replication through autophagy, require investigation. Moreover, a growing number of studies confirm that interferon-stimulated gene (ISG) products have an antiviral function in innate immunity by regulating and manipulating autophagy [70]. This is consistent with the results of our earlier work [43], where enrichment analyses showed that CAstV infection caused increased expression of mostly just ISGs involved in various signaling pathways, mainly the RIG-I-like signaling pathway. This is in line with the results of our KEGG functional analysis, as well as the results of our previous work in which we demonstrated that CAstV infection caused the increased expression of mainly ISGs involved in various signaling pathways, primarily the RIG-I-like signaling pathway.

In this study, another enriched pathway in KEGG functional analysis was the Wnt signaling pathway. Interaction of the Wnt/β-catenin signaling pathway has so far been described for some human and animal viruses, including human immunodeficiency virus (HIV), herpes simplex virus 1 (HSV1), hepatitis B virus (HBV), influenza virus (IAV), bovine parainfluenza virus type 3 (BPIV3), porcine reproductive and respiratory syndrome virus (PRRSV), hepatitis C virus (HCV), influenza virus (IAV), porcine circovirus type P1, and bovine herpesvirus type 1 (BoHV1). The action of these viruses is twofold—some of them activate the Wnt/β-catenin signaling pathway, while others inhibit it through different mechanisms. Its role in avian virus infection remains very poorly understood with publications stating that the Wnt/β-catenin pathway was activated in ALV-J infection of chickens [71]. The Wnt pathway was also regulated after HAstV-1 infection of Caco-2 cells 8hpi [67].

FoxO proteins are key regulators of gene expression in apoptosis, cell cycle progression and resistance to oxidative stress. These cellular functions may play a role in virus–host interactions in innate immunity. It has been proven that in Sendai virus infection, FoxO1 acts as a negative regulator of RIG-I-triggered signaling, promotes viral replication and decreases type I IFN production. Japanese encephalitis virus causes a decrease in FoxO protein expression, which induces apoptosis. In addition, FoxO protein is involved in the regulation of innate immunity and antiviral mechanism in Coxsackievirus B3 infection [72].

The expression of ErbB receptors on epithelial cells plays an important role in pathogen contact, as they are the main point of contact with pathogens. ErbB receptors undergo endocytosis during their normal life cycle, but their ligation with the pathogen, as well as their signaling cascades, can somehow help microorganisms enter the cells. Such “takeover” of ErbB signaling pathways by pathogens leads to prolonged host cell survival and affects the immune response. It has been proven that viruses, including hepatitis C virus (HCV), Epstein–Barr virus (EBV) and human papillomavirus (HPV), use ErbB receptors for self-propagation [73].

Viral infection disrupts the normal functioning of the cell to optimize viral replication and virion production. One such change is the reconfiguration and reorganization of cellular actin, which is essential at every stage of the virus life cycle, from the moment it enters the host cell [74]. Viruses, as they move through cells, also force changes in the actin cytoskeleton. Such changes can be identified by specialized cells. A tyrosine kinase called focal adhesion kinase (FAK) is involved in relaying such signals. Its important role has been described in herpes simplex virus entry and in rabies virus infection. It also has important functions in the endosomal transport of influenza A viruses, as well as promoting their replication [75]. This study enriched miRNA targets in focal adhesion during CAstV infection. This result suggests that FAK may play an important role in CAstV entry. However, additional studies are needed to investigate the mechanisms underlying these observations.

Since viruses require the host’s transcription and translation mechanisms to reconstruct their genome and its associated proteins, viruses modulate normal cellular pathways to facilitate this process. mTOR is an evolutionarily conserved serine/threonine kinase that regulates the mTORC1 and mTORC2 protein complexes associated with cell growth. mTOR signaling has been proven to be essential for viral translation. Both DNA viruses, e.g., adenoviruses, cytomegalovirus, and herpes viruses, as well as RNA viruses, e.g., Middle East respiratory syndrome coronavirus [MERS-CoV], influenza, HIV, West Nile virus [WNV] and Zika virus [ZIKV]), modulate the mTOR pathway by activating phosphoinositide 3-kinase (PI3K), Akt, or mTOR alone. mTOR inhibition not only inhibits viral protein synthesis, but also disrupts viral-mediated transcription [76].

Research on the role of miRNAs in viral infections is needed to provide insight into the function of miRNAs in intracellular communication and induction of antiviral responses. Learning about the mechanisms that may be regulated by miRNAs during infection will expand current knowledge of host–pathogen interactions. Such regulation is multifaceted, and although host miRNAs can positively regulate antiviral responses, viruses have also been shown to influence miRNA expression to favor viral infection [33,77].

## 4. Materials and Methods

### 4.1. Virus and Animals

The CAstV strain PL/G059/2014 (GenBank accession No. JF414802), associated with “white chicks syndrome”, was propagated on embryonated specific pathogen-free (SPF) chicken eggs, as described previously [18]. Internal organs of PL/G059/2014-infected chicken embryo homogenate were stored at −70 °C until further use.

SPF chicken embryos were obtained from VALO BioMedia (Osterholz-Scharmbeck, Germany) and hatched and housed in isolators until use.

### 4.2. Animal Experiments

The infection experiment was performed as described previously [43]. Briefly, five 3-week-old SPF chickens were infected with CAstV, while five not infected SPF chickens were used as a negative control (10 chickens in total). Cloacal swabs were collected from infected chicks on the second and fourth days post-inoculation (dpi) to check for the presence of virus, as described in [78]. Birds were necropsied at 4 dpi and the spleen samples were collected and stored in −70 °C for further procedures.

All chickens used in this study were housed in a BSL-3 experimental facility during the trial. Feed and water were supplied ad libitum. The chickens were observed daily for the duration of the experiment.

All animal experiments were approved and carried out in accordance with the guidelines set forth by the Local Ethics Commission.

### 4.3. Tissue Homogenization and Total RNA Isolation

Spleen samples were homogenized in RLT buffer containing β-mercaptoethanol, by use of an MP FastPrep-24 Tissue and Cell Homogenizer. Total RNA was extracted from 200 μL of tissue homogenate using the mirVANA miRNA Isolation Kit (Life Technologies, Thermo Fisher Scientific, Carlsbad, CA, USA).

### 4.4. RNA Quantification and Quality Control

RNA concentrations were measured using a Qubit RNA Assay Kit and a Qubit Fluorometer (Invitrogen, Carlsbad, CA, USA). The quality of the RNA was evaluated by the Bioanalyzer 2100 instrument (Agilent Technologies, Santa Clara, CA, USA) using a small RNA chip (Agilent Technologies, Santa Clara, CA, USA) according to the manufacturer’s protocol for further cDNA synthesis and sequencing. The RNA integrity number (RIN) of each sample exceeded the threshold of 7.

### 4.5. Small RNA Sequencing

RNA samples were used to construct cDNA libraries using Illumina TruSeq Small RNA Library Preparation Kit following the manufacturer’s instructions in an external commercial service (Macrogen Inc., Seoul, Republic of Korea). The libraries were sequenced on an Illumina HiSeq 2500 Rapid Run Mode sequencer using SBS (Sequencing by Synthesis) reagents. Base calling was conducted through an integrated primary analysis software called Real Time Analysis (RTA) and the output of RTA was converted to FastQ format data with Illumina bcl2fastq (Illumina, San Diego, CA, USA).

### 4.6. Raw Data Analysis

The quality control of raw sequence reads was performed using FastQC v0.11.7 sofware [79]. Adapter sequences were trimmed off the raw sequence reads using Cutadapt 1.16 [80]. The trimmed reads were used as processed reads to analyze long targets (≥50 bp). Unique clustered reads were sequentially aligned to the reference genome using miRBase v21, and the non-coding RNA database, RNAcentral 10.0, to classify known miRNAs. Novel miRNA prediction was performed by miRDeep2 [81,82,83]. The read counts for each miRNA were extracted from mapped miRNAs to report the abundance of each miRNA. Differentially expressed miRNAs were determined by comparing each miRNA across conditions (virus infected vs. non-infected) using statistical methods with DESeq2 in Bioconductor/R [84,85]. Only miRNAs that have changed their expression of adjusted *p*-value < 0.05 were classified as differentially expressed miRNAs. For target genes prediction of the differentially expressed miRNAs, the miRDB—an online database for prediction of functional microRNA targets was used [86]. In this study, we were only interested in CAstV-induced differential expression of chicken miRNA, and therefore the target genes were searched against chicken genome. We chose targets with prediction scores above 80. The target genes were dedicated to perform functional enrichment analysis on the target. The targets were subjected to enrichment analyses of Gene Ontology (GO) and the Kyoto Encyclopedia of Genes and Genomes (KEGG) pathways using DAVID 8.6 [87,88,89]. Fisher’s exact test was used for the enrichment analyses. GO terms and pathways with a *p*-value ≤ 0.05 were considered as significantly enriched.

### 4.7. Accession Number

The raw sequencing data obtained in this study were submitted to the Sequence Read Archive (SRA) database under accession number SUB13822657. The sequences were released on 8 September 2023.

### 4.8. Real-Time PCR Validation of RNA Reads

cDNA was synthesized using TaqMan miRNA Reverse Transcription Kit according to manufactures instructions and using Inventoried probes purchased in Life Technology (gga-miR-2954 Assay ID—243071, gga-miR-3533—Assay ID—242980_mat, gga-miR-7—Assay ID—000386, gga-miR-1664-5p Assay ID—007313_mat). Briefly the procedure involved poly(A) tail 3′ end generation followed by adaptor ligation at miRNA’s 5′ end. Then, the universal primers were used to perform reverse transcriptase reaction for all of the miRNAs. Real-time PCR was carried out using TaqMan^®^ Universal Master Mix II, and miRNA expression was evaluated in the 7500 fast system Real-Time PCR System (Applied Biosystems, Foster City, CA, USA). The LSM6 gene (Life Technology, Assay ID—Gg03321755_m1) was used as an endogenous control to normalize miRNA expression. The samples were run in duplicate. The 2^−ΔΔCt^ method was used to calculate the expression level of four selected miRNAs, which were then expressed as the fold change in gene expression.

### 4.9. Statistical Analysis of qRT-PCR

Statistical analysis was performed in Statistica 13.3 software (TIBCO Software Inc. Palo Alto, CA, USA. The Kruskal–Wallis or the Mann–Whitney test was used to evaluate the differences between continuous variables.

## 5. Conclusions

Our studies identified miRNAs that were differentially expressed during CAstV infection in chickens. The applied bioinformatics analysis made it possible to select the molecular pathways activated after the virus enters the host cells. The target genes for these specific miRNAs were predicted to include genes mainly involved in the regulation of cellular processes and immune responses. These data will be helpful in elucidating the mechanism underlying CAstV infection as well as the host-viral response in relation to miRNA expression. However, further studies should be undertaken to determine miRNA expression in larger study groups and at different time intervals after virus entry. This would provide an in-depth understanding of the molecular pathogenesis in chickens during CAstV infection.

## Figures and Tables

**Figure 1 ijms-24-15128-f001:**
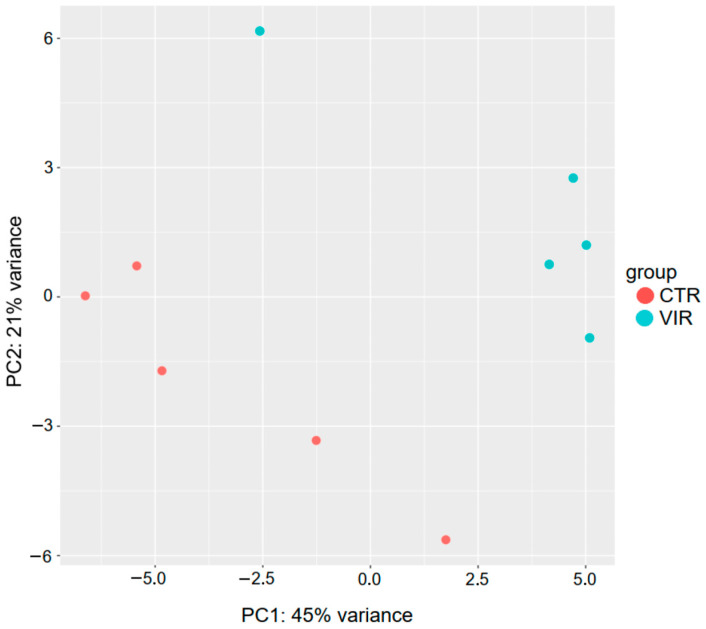
Principal-component analysis (PCA) of the RNA-seq samples. CAstV-infected (VIR) and control (CTR) groups are represented by different colors as indicated by the legend provided within the graph. Each dot represents a biological replicate of an RNA-seq sample.

**Figure 2 ijms-24-15128-f002:**
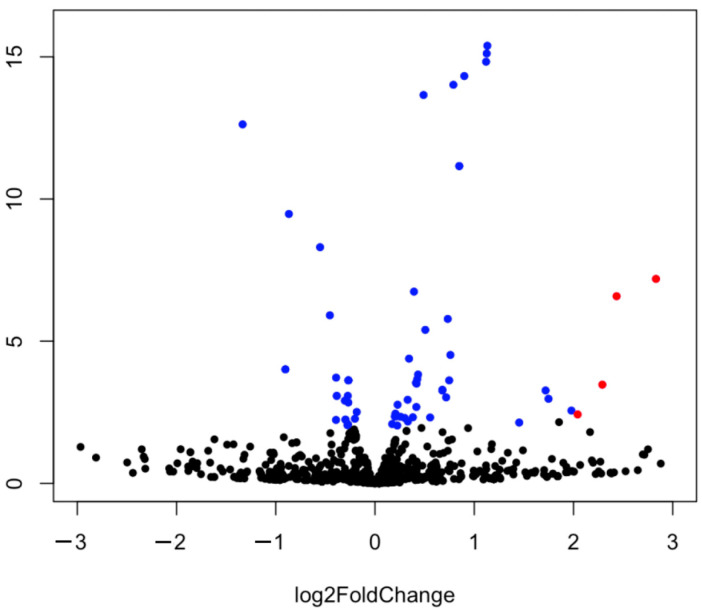
Volcano plot analysis of differentially expressed miRNA between the control and CAstV-infected groups, upregulated (right side) or downregulated (left side). miRNAs that changed their expression significantly (*p* < 0.05) are shown in blue, or in red if that also passed the 2 fold-change threshold (2Log FC > 2; *p* < 0.05). Non-significantly expressed genes are shown in black.

**Figure 3 ijms-24-15128-f003:**
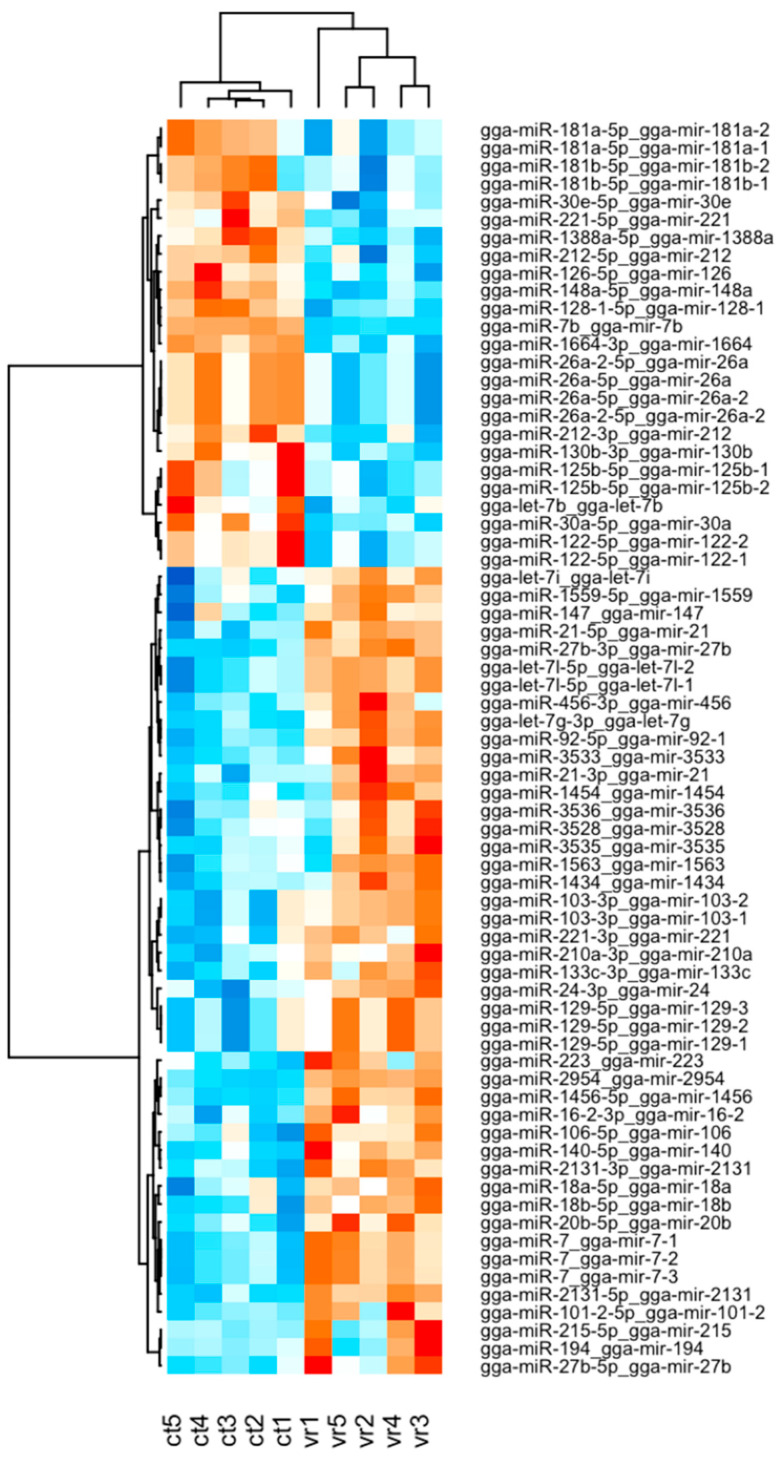
Heat map analysis used to classify miRNA expression (*p* < 0.05,  |log2FC | > 1) patterns under different experimental condition {virus-infected (vr) vs. non-infected (ct)}. miRNAs with similar expression patterns were clustered into groups in the heat map. Intensity of color indicates miRNA expression levels. Red/orange represents miRNAs with high levels of expression and blue represents miRNAs with low levels of expression.

**Figure 4 ijms-24-15128-f004:**
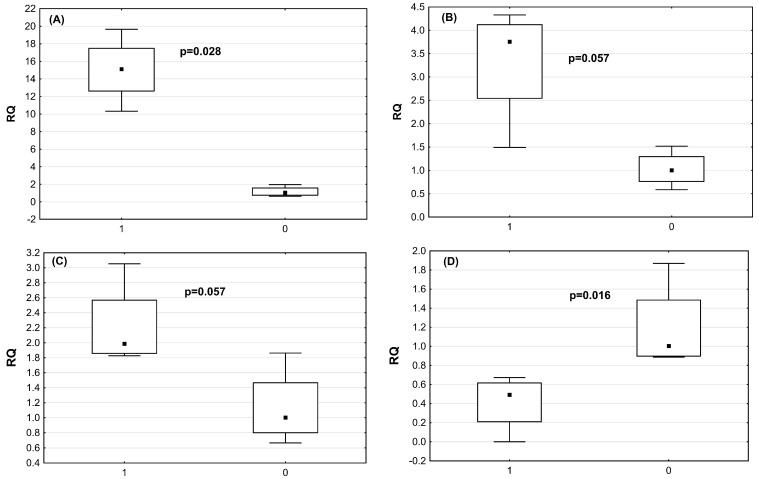
miR-2954 (**A**), miR-3533 (**B**), miR-7 (**C**), and miR-1664 (**D**) expression in CAstV-infected chickens (1) in relation to non-infected chickens (0), as verified by qRT-PCR.

**Table 1 ijms-24-15128-t001:** Ten most down- and upregulated DE miRNAs in CAstV-infected chickens.

	Base Mean	log 2 Fold Change	*p*-Value	p adj	reg
miR-7b	164.4544	−9.9228	4.88002383607237 × 10^−27^	8.05203932951941 × 10^−25^	down
miR-1664-3p	147.0916	−1.3340	2.37707360550244 × 10^−13^	8.71593655350895 × 10^−12^	down
miR-122-5p	127.9187	−0.9030	9.76991651583061 × 10^−5^	0.00140177063053222	down
miR-128-1-5p	243.0959	−0.8677	3.35882982217558 × 10^−10^	9.23678201098283 × 10^−9^	down
miR-148a-5p	1892.0338	−0.5535	4.95174600295029 × 10^−9^	1.256981677672 × 10^−7^	down
miR-1388a-5p	322,547.0625	−0.4547	1.22866160658981 × 10^−6^	2.38504900102727 × 10^−5^	down
let-7b	11,727.14297	−0.3930	0.00587255446460657	0.0307609995765106	down
miR-212-3p	1008.2254	−0.3917	0.000191195025360761	0.00252377433476205	down
miR-181a	361,275.1404	−0.3852	0.000837453238149873	0.00674341931614952	down
miR-212-5p	919.3084	−0.3038	0.00123092502988673	0.00902678355250267	down
miR-7	728.0242	1.1319	4.08074456514255 × 10^−16^	4.48881902165681	up
miR-3536	435.0424	1.4528	0.0072657815031363	0.0368878137851535	up
miR-1563	404.4242	1.7188	0.000539888145993531	0.00470226549287548	up
miR-215-5p	883.0175	1.7482	0.00106070451263859	0.00814029044583105	up
miR-3528	168.3393	1.9790	0.00275325105712132	0.0185423030377558	up
miR-194	46.1090	2.0418	0.00376268623521676	0.023878585723491	up
miR-3535	199.8722	2.2906	0.00033749035146587	0.0032756416465805	up
miR-1434	31.2215	2.4343	2.61622382369712 × 10^−7^	5.39596163637531 × 10^−6^	up
miR-3533	14.7673	2.8310	6.44514082042924 × 10^−8^	1.51921176481546 × 10^−6^	up
miR-2954	13,896.9314	3.2233	5.95623585116667 × 10^−102^	1.965557830885 × 10^−99^	up

**Table 2 ijms-24-15128-t002:** Top 10 GO terms for biological process ranked by *p*-value following analysis of DE miRNA target genes.

Description	*p*-Value	Fold Enrichment	FDR
GO:0006357~regulation of transcription from RNA polymerase II promoter	8.28583322838162 × 10^−8^	1.3399	3.7253106194803763 × 10^−4^
GO:0035556~intracellular signal transduction	3.545900675891955 × 10^−7^	1.6623	7.971184719405115 × 10^−4^
GO:0043087~regulation of GTPase activity	1.8096098121506414 × 10^−5^	2.2200	0.020594082232785374
GO:0006468~protein phosphorylation	1.8322137217780583 × 10^−5^	1.5982	0.020594082232785374
GO:0007411~axon guidance	3.0001284374567666 × 10^−5^	1.8560	0.026977154909611247
GO:0000209~protein polyubiquitination	1.2557755955090277 × 10^−4^	1.9001	0.0864692158518967
GO:0018105~peptidyl-serine phosphorylation	1.3462733784770392 × 10^−4^	1.7754	0.0864692158518967
GO:0007399~nervous system development	2.4972724679447954 × 10^−4^	1.6531	0.1403467126984975
GO:0051965~positive regulation of synapse assembly	4.989835284296587 × 10^−4^	2.4461	0.2492699937577495
GO:0071526~semaphorin-plexin signaling pathway	9.198871146956833 × 10^−4^	2.8591	0.4135812467671792

**Table 3 ijms-24-15128-t003:** Top 10 GO terms for molecular function ranked by *p*-value following analysis of DE miRNA target genes.

Description	*p*-Value	Fold Enrichment	FDR
GO:0004674~protein serine/threonine kinase activity	3.0866587938833846 × 10^−7^	1.6959	4.2256358888263535 × 10^−4^
GO:0000978~RNA polymerase II core promoter proximal region sequence-specific DNA binding	1.1901488065126979 × 10^−6^	1.3545	8.146568580579417 × 10^−4^
GO:0046872~metal ion binding	1.0696663302522198 × 10^−4^	1.2135	0.03864123656365064
GO:0003700~transcription factor activity, sequence-specific DNA binding	1.1290353999605739 × 10^−4^	1.4974	0.03864123656365064
GO:0042802~identical protein binding	2.2739749560208233 × 10^−4^	1.2919	0.06226143429585014
GO:0001227~transcriptional repressor activity, RNA polymerase II transcription regulatory region sequence-specific binding	4.552443546848259 × 10^−4^	1.5698	0.10387158692725443
GO:0005085~guanyl-nucleotide exchange factor activity	6.272465550624852 × 10^−4^	1.6860	0.12170592865052825
GO:0004712~protein serine/threonine/tyrosine kinase activity	7.112106860512972 × 10^−4^	1.4384	0.12170592865052825
GO:0005096~GTPase activator activity	9.016902518531111 × 10^−4^	1.6080	0.1325395104834417
GO:0003682~chromatin binding	0.001034656718005266	1.4259	0.1325395104834417

**Table 4 ijms-24-15128-t004:** Top 10 GO terms for cellular component ranked by *p*-value following analysis of DE miRNA target genes.

Description	*p*-Value	Fold Enrichment	FDR
GO:0005634~nucleus	3.8040482560238504 × 10^−6^	1.1461	0.0029591683447036292
GO:0098978~glutamatergic synapsę	7.6071165673615145 × 10^−6^	1.7482	0.0029591683447036292
GO:0005829~cytosol	2.433178408974373 × 10^−5^	1.1812	0.006310042673940207
GO:0005794~Golgi apparatus	4.615516814134699 × 10^−5^	1.3954	0.007716152286538233
GO:0005886~plasma membranę	4.958966765127399 × 10^−5^	1.1752	0.007716152286538233
GO:0005654~nucleoplasm	4.6747909613586004 × 10^−4^	1.1800	0.060616456132283184
GO:0005942~phosphatidylinositol 3-kinase complex	8.362418083447217 × 10^−4^	2.6013	0.09294230384174193
GO:0043197~dendritic spine	0.0010418476883945682	1.9510	0.10131968769637176
GO:0016020~membrane	0.0025990445446487298	1.2330	0.22467296174852353
GO:0005737~cytoplasm	0.004274525839628792	1.0893	0.3115246085292393

**Table 5 ijms-24-15128-t005:** KEGG pathways enrichment for DE miRNA target genes in chickens infected with CAstV at 4 dpi.

Description	*p*-Value	Fold Enrichment	FDR
gga04010:MAPK signaling pathway	6.9619020229201035 × 10^−9^	1.5804	1.1278281277130568 × 10^−6^
gga04070:Phosphatidylinositol signaling system	1.4035027273706892 × 10^−6^	1.8709	1.1368372091702583 × 10^−4^
gga04144:Endocytosis	3.5017161152942025 × 10^−6^	1.4888	1.8909267022588694 × 10^−4^
gga04520:Adherens junction	1.0685138134129477 × 10^−5^	1.7929	4.327480944322438 × 10^−4^
gga04140:Autophagy-animal	4.6602982670722855 × 10^−5^	1.5888	0.0015099366385314204
gga04310:Wnt signaling pathway	2.5808668477564584 × 10^−4^	1.4834	0.006968340488942438
gga04068:FoxO signaling pathway	0.001287555905712632	1.4785	0.026319355164882662
gga04012:ErbB signaling pathway	0.0012997212427102549	1.6218	0.026319355164882662
gga00562:Inositol phosphate metabolism	0.0018404595627396826	1.6353	0.03312827212931429
gga04510:Focal adhesion	0.004997193963046481	1.3189	0.07461497775631122
gga04910:Insulin signaling pathway	0.005066449106910021	1.4165	0.07461497775631122
gga04150:mTOR signaling pathway	0.008692914782026601	1.3490	0.11735434955735911
gga04810:Regulation of actin cytoskeleton	0.011161569231401824	1.2753	0.12952353934521796
gga04912:GnRH signaling pathway	0.01119339228909291	1.4628	0.12952353934521796
gga04114:Oocyte meiosis	0.01204386517678131	1.4082	0.13007374390923815
gga03250:Viral life cycle—HIV-1	0.01976756804864089	1.5644	0.20014662649248902
gga04261:Adrenergic signaling in cardiomyocytes	0.02273549527770102	1.3176	0.21665589617573913
gga04625:C-type lectin receptor signaling pathway	0.0242064397986349	1.3955	0.2178579581877141
gga04216:Ferroptosis	0.030580226017975658	1.6819	0.26073666394273987
gga00564:Glycerophospholipid metabolism	0.03379259593242035	1.3510	0.27372002705260484

**Table 6 ijms-24-15128-t006:** miR-1664, miR-2954, miR-3533 and miR-7 expression in CAstV-infected and non-infected chickens, as verified by RT-PCR.

miRNA	Astrovirus-Infected Chickens RQ Median (Mean, SD)	Non-Infected Chickens RQ Median (Mean, SD)	*p*-Value
miR-2954	15.1 (15.0; +/−3.8)	1.04 (1.17; +/−0.58)	*p* = 0.028
miR-3533	3.75 (3.33; +/−0.3)	1.0 (1.0; +/−38)	*p* = 0.57
miR-7	1.99 (2.21; =/−0.57)	1.0 (1.14; +/−0.5)	*p* = 0.57
miR-1664	0.5 (0.41; +/−0.29)	1.1 (1.24; +/−1.86)	*p* = 0.016

## Data Availability

The raw data generated in RNA-seq study were submitted to the SRA database (https://www.ncbi.nlm.nih.gov/sra) (accessed on 8 September 2023) under accession number SUB13822657 (release date: 8 September 2023).

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
