# Peer review of "miRNA Expression Signatures Induced by Chicken Astrovirus Infection in Chickens"

_ijms, 2023, doi:10.3390/ijms242015128_

Round 1

Reviewer 1 Report

In the study with the title “miRNA expression signatures induced by chicken astrovirus infection in chickens” the authors perform a small RNA sequencing experiment in order to find differentially expressed miRNAs in chickens after astrovirus infection. The authors assess the regulatory potential of these differentially expressed miRNAs in silico and validate the levels of significantly deregulated miRNAs with qPCR. Even though the aim of the study is achieved by proposing specific miRNAs as significant, some major issues exist in the Methods, Results, and Discussion section of the study:

·         The results about miR-7b seem contradictory, as it appears downregulated in NGS data but relatively upregulated in the qPCR experiments. How can this be explained? Therefore, the part in the manuscript where the authors state that the NGS data we validated seems bizarre.

·         The Discussion section seems extremely detailed and expanded. As a result, the significance of the findings is not clarified. Therefore, it is highly recommended to limit this part, exclude repetitive information referring to results, and provide a more compact Discussion.

·         Figures 1, 3, 4: The quality of the figures is poor and the numbers are relatively small. Moreover, it is essential to provide the corresponding abbreviations of the two groups studied in the legends.

·         What was the RNA input mass that was used for cDNA synthesis? Moreover, please provide the primers that were used for the amplification of miRNAs with qPCR. Did the authors check the limitations for the use of 2-ΔΔCt?

·         It is suggested to show miRNAs in the same order in Figure 4 and Table 6. Otherwise, it may lead the reader to confusion.

Author Response

In the study with the title “miRNA expression signatures induced by chicken astrovirus infection in chickens” the authors perform a small RNA sequencing experiment in order to find differentially expressed miRNAs in chickens after astrovirus infection. The authors assess the regulatory potential of these differentially expressed miRNAs in silico and validate the levels of significantly deregulated miRNAs with qPCR. Even though the aim of the study is achieved by proposing specific miRNAs as significant, some major issues exist in the Methods, Results, and Discussion section of the study:

OUR RESPONSE: We thank Reviewer 1 for the comments concerning our manuscript. Please find the detailed responses below.

1. The results about miR-7b seem contradictory, as it appears downregulated in NGS data but relatively upregulated in the qPCR experiments. How can this be explained? Therefore, the part in the manuscript where the authors state that the NGS data we validated seems bizarre.

OUR RESPONSE: We thank Reviewer 1 for this comment. Contradictory results of Real-Time PCR and NGS are due to wrong caption under the Figure showing overexpression of miR-7 (NOT miR of -7b as stated incorrectly). We changed this mistyped error in text in lines: 276, 299, 301.

2. The Discussion section seems extremely detailed and expanded. As a result, the significance of the findings is not clarified. Therefore, it is highly recommended to limit this part, exclude repetitive information referring to results, and provide a more compact Discussion.

OUR RESPONSE: We thank Reviewer 1 for this comment. Where possible, we have shortened the Discussion by removing passages that are partial repetitions of the Introduction and Results. However, we could not shorten this part very radically, because the topic is quite complex and requires a detailed approach to be understood by the Reader. Deleted fragment in lines: 304-326.

3. Figures 1, 3, 4: The quality of the figures is poor and the numbers are relatively small. Moreover, it is essential to provide the corresponding abbreviations of the two groups studied in the legends.

OUR RESPONSE: We thank Reviewer 1 for this comment. We enlarged some of the figures and descriptions. We hope they are now readable. 

4. What was the RNA input mass that was used for cDNA synthesis? Moreover, please provide the primers that were used for the amplification of miRNAs with qPCR. Did the authors check the limitations for the use of 2-ΔΔCt?

OUR RESPONSE: We thank Reviewer 1 for this comment. RNA input, as well as its quality and quantity was assessed and RNA that met strict criteria were qualified to further analysis (RIN>7). A total of around 10 ng was used to cDNA synthesis (according to manufacturer’s recommendations).

We performed 2-ΔΔCt analysis since we had Life Technology software tool (widely acknowledged) that enabled such an analysis. To ensure that 2-ΔΔCt method for calculation is reliable (to overcome its limitation and to have high enough amplification efficiency) we always performed, before starting the experiment, as follows: firstly non regulated reference gene expression analysis (preferably at least two genes LSM and Usb1) to find the suitable endogenous control with stable expression within all studied groups. Secondly multiple dilutions of cDNA sample to determine optimal concentration were carried out. Also every sample were amplified in duplicate to ensure repeatability of the results.

Apart the high sample quality, the potential negative impact of other factors on amplification efficiency (like master mix performance or primer specificity and annealing temperature) were minimalized thanks to use of standardized Real-Time kits/reagent/primer purchased commercially.

The Assay ID of primer used for Real-Time PCR analysis are added to the manuscript in lines 162-163 and 168-169.

gga-miR-7 - Assay ID - 000386 Life Technology

gga-miR-3533 - Assay ID - 242980_mat Life Technology

gga-miR-1664-5p Assay ID 007313_mat Life Technology

gga-miR-2954 Assay ID 243071_mat Life Technology

Lsm5 Assay ID Gg03328933_m1 Life Technology

We also provide for the Reviewer's information the ID number of Usb1: Assay ID Gg07181230_m1 Life Technology

5. It is suggested to show miRNAs in the same order in Figure 4 and Table 6. Otherwise, it may lead the reader to confusion.

OUR RESPONSE: We thank Reviewer 1 for this comment. We have included this change in the manuscript.

Reviewer 2 Report

Dear Authors

Thank you for a neat, sound and concise manuscript.  See below comments/ recommendations:

Text in Figures 3 and 4 is not visible.  Needs to be larger in size

1. Lines 35 to 39:  Split this single sentence into two or more separate ones.  There is too much of information in this sentence.

2. Line 66:  Correct "Jopling at all"

3.  Line 79:  Incorrect spelling "scarce"

4.  Line 86:  Consider sentence; "pathways engages in potentially vital for.." -  is there a word missing?

5.  Line 92:  Consider "PL/G059/2014-infected" chicken embryos

6.  Line137:  "conditions" are stated, but no description or list of the different conditions are given.  Cross reference with Figure 3, in which the heat map is used as a classification tool relative to the different "conditions"

7.  Line 276:  Add a reference for global incidence statement

8.  Line 306:  italicise Salmonella spp.

9.  Line 317 to 320: Rephrase sentence

The manuscript was easy to read.  Minor grammar and sentence structure edits have been highlighted

Author Response

We thank Reviewer 2 for the positive comments concerning our manuscript.

1. Text in Figures 3 and 4 is not visible.  Needs to be larger in size

OUR RESPONSE: We thank Reviewer 2 for this comment. We have included this change in manuscript.

2. Lines 35 to 39:  Split this single sentence into two or more separate ones.  There is too much of information in this sentence.

OUR RESPONSE: We thank Reviewer 2 for this comment. We have included this change in text in lines: 39-45.

3. Line 66:  Correct "Jopling at all"

OUR RESPONSE: We thank Reviewer 2 for this comment. We have included this change in text in line 72.

4. Line 79:  Incorrect spelling "scarce"

OUR RESPONSE: We thank Reviewer 2 for this comment. We have included this change in text in line 85.

5. Line 86:  Consider sentence; "pathways engages in potentially vital for.." -  is there a word missing?

OUR RESPONSE: We thank Reviewer 2 for this comment. The sentence reflects our intentions.

6. Line 92:  Consider "PL/G059/2014-infected" chicken embryos

OUR RESPONSE: We thank Reviewer 2 for this comment. We have included this change in text in line 98.

7. Line137:  "conditions" are stated, but no description or list of the different conditions are given.  Cross reference with Figure 3, in which the heat map is used as a classification tool relative to the different "conditions"

OUR RESPONSE: We thank Reviewer 2 for this comment. We have included this change in text in lines: 143 and 232.

8. Line 276:  Add a reference for global incidence statement

OUR RESPONSE: We thank Reviewer 2 for this comment. This passage has been removed. Following comments from other Reviewers that the Discussion is too elaborate, we have decided to remove this excerpt, as it was partly a repetition of an excerpt from the Introduction.

9. Line 306:  italicise Salmonella spp.

OUR RESPONSE: We thank Reviewer 2 for this comment. We have included this change in text in line 340.

10. Line 317 to 320: Rephrase sentence

OUR RESPONSE: We thank Reviewer 2 for this comment. We have included this change in text in lines: 355-358.

Reviewer 3 Report

Dear authors,

in your work you investigated, by NGS, miRNA expression in SPF chickens after CastV inoculation.  Abstract, Introduction and Material and Methods are detailed, give an insight into the topic of investigation and describe the method of conducting experiments in detail. Results are also well presented. Possible shortcomings are really long Discussion i (is it possible to shorten it?) and (as also stated by you in Conclusion) small number of chickens used in experiment as well as the duration of it.

Specific comments:

Please uniform writing of DE miRNA (or DEmiRNA) (L188 and 201)

L 31,61,156,157,158,440 -spacing in the sentences

L38 - hatchery - you mean "to decrease hatching"?

L 97-altogether 10 SPF chickens were used in this experiment? total duration of the experiment was 4 days (after inoculation)?

L109 - which tissue samples, spleen? 

L137 sentence needs revision (using-using)

L182 what do you mean by "different treatment" - control and inoculation?

L309 here you mention "we detected" - please provide reference

L340 please remove "are no also"

L 348 abbreviation IBV stands for infectious bronchitis virus?

Author Response

We thank Reviewer 3 for the positive comments concerning our manuscript. Where possible, we have shortened the Discussion by removing passages that are partial repetitions of the Introduction and Results. However, we could not shorten this part very radically, because the topic is quite complex and requires a detailed approach to be understood by the Reader. This study provided broad insight into the CAstV-induced host response and a basis for further studies that may clarify the interactions between virus and host. Although this study had some limitations, such as using small study groups and a single time point, it, nonetheless, provided a global analysis of host miRNA expression changes that occur during CAstV infection in vivo, new information about miRNAs in chickens, and a strong basis for further studies.

Specific comments:

1. Please uniform writing of DE miRNA (or DEmiRNA) (L188 and 201)

OUR RESPONSE: We thank Reviewer 3 for this comment. We have included these changes in text in line 210.

2. L 31,61,156,157,158,440 -spacing in the sentences

OUR RESPONSE: We thank Reviewer 3 for this comment. We have included these changes in the manuscript.

3. L38 - hatchery - you mean "to decrease hatching"?

OUR RESPONSE: We thank Reviewer 3 for this comment. We have included this change in text in line 42.

4. L 97-altogether 10 SPF chickens were used in this experiment? total duration of the experiment was 4 days (after inoculation)?

OUR RESPONSE: We thank Reviewer 3 for this comment. Yes, there were ten chickens in total and the experiment duration was 4 days.  We have included this change in text in line 105.

5. L109 - which tissue samples, spleen? 

OUR RESPONSE: We thank Reviewer 3 for this comment. We have included this change in text in line 115.

6. L137 sentence needs revision (using-using)

OUR RESPONSE: We thank Reviewer 3 for this comment. We have included this change in text in lines 143-144.

7. L182 what do you mean by "different treatment" - control and inoculation?

OUR RESPONSE: We thank Reviewer 3 for this comment. Yes, we mean virus-infected vs. Non-infected chickens. We have included this change in text in lines: 143, 191-192 and 232.

8. L309 here you mention "we detected" - please provide reference

OUR RESPONSE: We thank Reviewer 3 for this comment. miR-2954 is one of the DE miRNAs we identified in this work. That is why we wrote „we detected”.

9. L340 please remove "are no also"

OUR RESPONSE: We thank Reviewer 3 for this comment. We have changed the construction of this sentence in text in lines 379-380.

10. L 348 abbreviation IBV stands for infectious bronchitis virus?

OUR RESPONSE: We thank Reviewer 3 for this comment. We have added the explanation of the abbreviation in the lines 344-345.

Round 2

Reviewer 1 Report

Previous comments have been properly addressed except comment 3. The quality of the Figures and the size of the text in Figures 1,3, and 4 could be greatly improved.

Author Response

OUR RESPONSE: We thank Reviewer 1 for this comment. We have changed the figures in accordance with the Reviewer's comment and we hope that now their quality and readability are satisfactory.